# Lost in the Averages: Evaluating record-specific MIAs against Machine Learning models

## Abstract

Record-specific Membership Inference Attacks (MIAs) are widely used to evaluate the propensity of a machine learning (ML) model to memorize an individual record and the privacy risk its release therefore poses. Record-specific MIAs are currently evaluated the same way ML models are: on a test set of models trained on data samples that were not seen during training ($D_{eval}$). A recent large body of literature has however shown that the main risk often comes from outliers, records that are statistically different from the rest of the dataset. In this work, we argue that the traditional evaluation setup for record-specific MIAs, which includes dataset sampling as a source of randomness, incorrectly captures the privacy risk. Indeed, what is an outlier is highly specific to particular data samples, and a record that is an outlier in the training dataset will not necessarily be one in the randomly sampled test datasets. We propose to use model randomness as the only source of randomness to evaluate record-level MIAs, a setup we call *model-seeded*. Across 10 combinations of models, datasets, and attacks for predictive and generative AI, we show the per-record risk estimates given by the traditional evaluation setup to substantially differ from ones given by the *model-seeded* setup which properly account for the increased risk posed by outliers. We show that across setups the traditional evaluation method leads to a substantial number of records to be incorrectly classified as low risk, emphasizing the inadequacy of the current setup to capture the record-level risk. We then a) provide evidence that the traditional setup is an average–across datasets–of the *model-seeded* risk, validating our use of model randomness to create evaluation models and b) show how relying on the traditional setup might conceal the existence of stronger attacks. The traditional setup would indeed strongly underestimate the risk posed by the strong Differential Privacy adversary. We believe our results to convincingly show the practice of randomizing datasets to evaluate record-specific MIAs to be incorrect. We then argue that relying on model randomness, an setup we call *model-seeded* evaluation, better captures the risk posed by outliers and should be used moving forward to evaluate record-level MIAs against machine learning models, both predictive and generative.

## 1 Introduction

Predictive and generative Machine Learning (ML) models are increasingly trained or fine-tuned by companies, governments, and academic researchers often using data that can be personal. Originally developed for aggregate data (Homer et al., 2008; Sankararaman et al., 2009), membership inference attacks (MIAs) have become the standard method to evaluate the privacy risks of ML models (Shokri et al., 2017; Carlini et al., 2022a). In particular, record-specific MIAs are used to evaluate the risk of an attacker being able to infer that a target record was a member of the released model's training set.

Record-level MIAs against ML models are typically instantiated as a binary meta-classifier, predicting membership for a given target record (Shokri et al., 2017; Carlini et al., 2022a; Stadler et al., 2022). The meta-classifier is trained using auxiliary data, and then evaluated on models trained on datasets randomly sampled from a large held-out data pool dedicated to evaluation. The target record is then added to a fraction of the datasets, typically to half of them to create a balanced evaluation setup.

Though using dataset sampling as a source of randomness is a natural extension of the ML evaluation pipeline to MIAs, we argue that–in light of recent evidence–this does not properly evaluate the privacy risk posed by an ML model release in practice. Recent research on MIAs against machine learning models has indeed shown the privacy risk to mostly lie with outliers, records that are statistically different from the rest of the dataset, being memorized. Stadler et al. (2022) and Meeus et al. (2023) have, for instance, shown how outlier records, often minorities, are vulnerable to attacks on synthetic data generators. (Carlini et al., 2022b) showed outliers to be vulnerable when attacking predictive ML models. They, and further work, showed how removing high-risk records causes the risk of other records to increase, emphasizing how the risk of a record is highly dependent on the dataset it is in and proposed to measure the overall privacy risk using TPR at low FPR.

**Contribution**. In this paper, we argue–in light of recent evidence–that the traditional evaluation setup for record-level MIAs, which includes dataset sampling as a source of randomness, incorrectly captures the privacy risk for outliers. Instead we propose to use model randomness as the only source of randomness to evaluate record-level MIAs, a setup we call *model-seeded*.

First, we describe the *traditional* MIA evaluation setup currently used in the literature with dataset sampling as a source of randomness. We then describe our proposed *model-seeded* evaluation setup which only uses randomness of the model - weight initialization and training randomness - as a source of randomness, which we argue allows it to capture the risk posed by outliers.

Second, we instantiate both evaluation setups across 10 combinations of models, datasets, and attacks for predictive and generative AI. We show the per-record risk estimates given by the traditional evaluation setup to substantially differ from ones given by the *model-seeded* setup which properly account for the increased risk posed by outliers. For instance, we find that 94% of high-risk records in the Adult dataset would be incorrectly considered as low-risk when using the traditional setup to evaluate a model trained using synthpop.

Third, we derive theoretical results which, combined with empirical evidence, strongly suggest that the risk calculated in the traditional setup is indeed an average of the risks specific to each dataset sampled for testing, as evaluated using the *model-seeded* setup. We argue these results validate our use of model randomness as the (only) source of randomness when evaluating record-specific MIAs against ML models.

Finally, we show that the traditional setup would strongly underestimate the risk posed by the strong Differential Privacy (DP) adversary. We instantiate an MIA attack by the very strong DP attacker with knowledge of the training dataset $D_{target}$. We show that this attack leads to drastically and significantly improved membership inference when evaluated in the model-seeded setup. Yet, the increased strength of the attacker has no measurable impact when evaluated in the traditional setup. Concerningly this show that the practice of using the traditional evaluation setup to evaluate new attacks risks concealing the existence and effectiveness of stronger attacks.

Taken together, we believe our results to convincingly show the practice of randomizing datasets to evaluate record-specific MIAs to be incorrect. We argue that relying on model randomness, a setup we call *model-seeded* evaluation, better captures the risk posed by outliers and should be used moving forward to evaluate record-level MIAs against machine learning models, both predictive and generative.

## 2 RELATED WORK

MIAs have become the standard method for auditing the privacy risk of ML models and synthetic data generators (Jagielski et al., 2020; Hayes et al., 2019; Steinke et al., 2024; Nasr et al., 2021). In particular, record-level MIAs are being used to evaluate the risk posed by the released model for each record and validate formal privacy guarantees (Stadler et al., 2022; Guépin et al., 2023; Ye et al., 2022; Houssiau et al., 2022).

While new techniques are continuously proposed (Leino & Fredrikson, 2020; Nasr et al., 2019; Yeom et al., 2018; Salem et al., 2018; Carlini et al., 2022a; Stadler et al., 2022), most rely on the shadow modeling technique introduced by Ateniese et al. (2015) and popularized by Shokri et al. (2017).

Song & Mittal (2020) introduce Mentr, an attack using a modified version of prediction entropy to infer membership, relying on the assumption that the expectation that a model's prediction entropy

will be higher on unseen samples. LiRA, introduced by Carlini et al. (2022a), combines shadow modelling and statistical testing to determine membership. Most recently, Zarifzadeh et al. (2024) introduced a new shadow model-based MIA that achieves high performances with fewer shadow models than previous attacks. As MIAs for ML models often rely on per-record predictions and loss values, they typically cannot be directly applied to synthetic data generators. Black-box attacks leveraging shadow models have thus been developed specifically for synthetic data. They rely on measuring the impact of the target record on the generated synthetic dataset by modelling the distributions of synthetic datasets generated with and without the target record. These include Stadler et al. (2022) and the state-of-the-art query-based attack which we use here (Houssiau et al., 2022).

Recent work has also shown privacy risk to vary across records, datasets, and classes (Tobaben et al., 2024; Yu et al., 2024), and the risk of a dataset to lie in most part with a small number of strongly memorized records (Feldman & Zhang, 2020). Outliers, in a general statistical sense, have been shown to be particularly vulnerable Meeus et al. (2023); Thudi et al. (2024); Stadler et al. (2022), and their risk to be highly dependent on other records in the dataset. As outliers are highly specific to the dataset they are in, this suggests that the risk of a record is also not absolute, but rather relative to the dataset it is contained in (Carlini et al., 2022b).

The standard evaluation setup for MIAs uses dataset sampling as a source of randomness. While not all works are explicit in their explanations of evaluation method, based on published work and available code, we have found that, to the best of our knowledge, record-specific risk is evaluated using dataset sampling as a source of randomness (Stadler et al., 2022; Carlini et al., 2022a;b; Guépin et al., 2023; Meeus et al., 2023; Houssiau et al., 2022).

**MIA performance evaluation.** Record-specific risk is traditionally evaluated using dataset sampling as a source of randomness. AUC or, alternatively, accuracy were the primary metrics used to measure the success of an MIA, usually with a balanced test set (Shokri et al., 2017; Choquette-Choo et al., 2021; Hayes et al., 2019). Further research demonstrated that privacy risk is not uniform across records, with some records shown to be much more vulnerable than average (Feldman & Zhang, 2020; Meeus et al., 2023; Carlini et al., 2022a; Stadler et al., 2022). Following these findings, metrics focusing on the most at-risk records were adopted as the de-facto standard in the MIA literature. This includes metrics focusing on the records identified a posteriori as being particularly at risk e.g. in synthetic datasets (Stadler et al., 2022; Meeus et al., 2023) and metrics measuring the risk the most vulnerable records across the entire dataset are exposed to, such as TPR (True Positive Rate) at low FPR (False Positive Rate). Recent evidence (Carlini et al., 2022b; Meeus et al., 2023) also suggests that at-risk records tend to be outliers, records that are different from other records in the dataset used to trained the model.

## 3 MIA EVALUATION SETUPS

### 3.1 NOTATION

**Records.** We consider a *tabular* record to consist of a finite set of $m$ attributes $x_i = (x_{i,1}, \cdots, x_{i,m}) \in \mathcal{F}_1 \times, \cdots, \times \mathcal{F}_m$. We denote by $\mathcal{F} = \mathcal{F}_1 \times, \cdots, \times \mathcal{F}_m$ the universe of possible records, and define $\mathcal{D}$ as a random distribution of records over $\mathcal{F}$.

We consider an *image* record to consist of a finite set of pixels, that can be written as a matrix $(X_{i,j}^r, X_{i,j}^g, X_{i,j}^b) \in [0, 255]^3$. For an image of size $n \times m$, we denote by $\mathcal{F} = ([0, 255]^3)^{n \times m}$ the universe of possible records, and define $\mathcal{D}$ as a random distribution of records over $\mathcal{F}$.

**Evaluation setup.** We denote $D_{target}$ the *target* dataset used to trained the target machine learning model $\mathcal{M}_{target}(D_{target})$ and $x_{target}$ the *target record* whose membership the attacker aims to infer. Note that for simplicity we use $\mathcal{M}_{target}$ as a shorthand for $\mathcal{M}_{target}(D_{target})$.

To accommodate–without loss of generality–the traditional and model-seeded setups, we denote by $D_{eval}$ the evaluation data pool available to the model developer to evaluate the privacy risk of releasing a trained model $\mathcal{M}_{target}$, including the dataset used to train $\mathcal{M}_{target}$ ($D_{target} \subset D_{eval}$).

We also use the term *evaluation models* for the models we use to evaluate the effectiveness of an MIA and *evaluation datasets* for the datasets they are trained on.

Finally, an auxiliary dataset $D_{aux}$ is used to train the MIA. We here first consider MIAs trained on $D_{aux}$, a dataset drawn from the same distribution as $D_{eval}$ (but strictly not overlapping), the most standard assumption in the literature (Shokri et al., 2017; Carlini et al., 2022a). In Sec. 7 specifically, we also develop an MIA where a very strong attacker, akin to the Differential Privacy attacker, has access to $D_{target}$ minus the membership information of $x_{target}$ as auxiliary dataset.

**Privacy risk.** We denote by $R^\phi_{setup}(x, D)$ the privacy risk for the record $x$ when evaluation datasets are sampled from $D$ and metric $\phi$ is used to calculate the performance score of the MIA. For metric $\phi$, we use ROC AUC, and report results using accuracy in Appendix G.

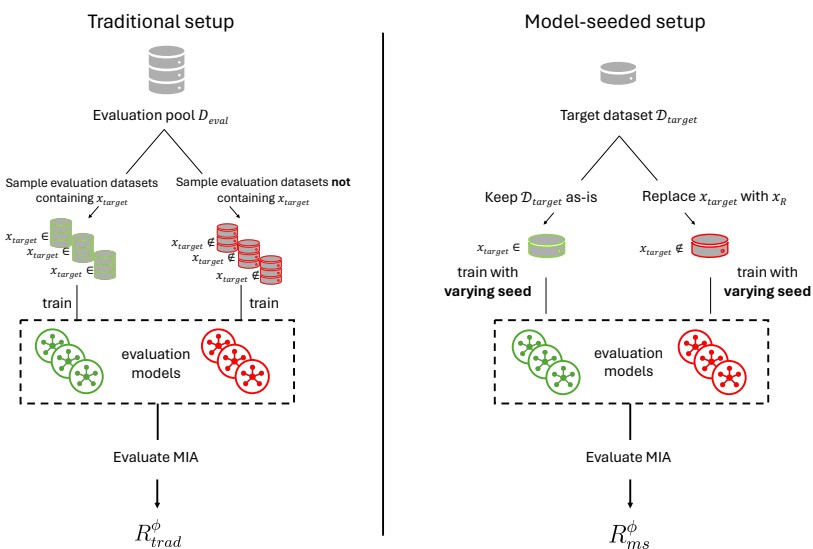

Figure 1: On the left: the traditional evaluation setup, sampling multiple datasets from the evaluation pool $D_{eval}$ and training an evaluation model on each. On the right: the model-seeded evaluation setup: the only evaluation datasets used are the full $\mathcal{D}_{target}$, in which $x_{target}$ is included, and $D_{target}$ in which $x_{target}$ is replaced by a reference record $x_R$ sampled from the evaluation pool $D_{eval}$.

## 3.2 TRADITIONAL EVALUATION SETUP

In the traditional evaluation setup, the MIA's ability to distinguish between models trained on a target record $x_{target}$ and models not trained on $x_{target}$ is evaluated by performing the MIA on multiple models, all trained on different data samples taken from the evaluation pool, where $x_{target}$ is included in exactly half of the samples. The source of randomness in this setup is thus the sampling of evaluation datasets, leading to a risk score aggregated over the sampled datasets. A diagram of this setup is presented on the left-hand side of Figure 1. We write $R^\phi_{trad}(x, D_{eval})$ (simplified as $R^\phi_{trad}$) for the risk estimated using the traditional setup using metric $\phi$.

## 3.3 MODEL-SEEDED EVALUATION SETUP

In the model-seeded MIA evaluation setup, we remove the randomness coming from the sampling of evaluation datasets, and use the the model randomness–weight initialization and training seed–of the evaluation models as the sole source of randomness. More specifically, we train $\frac{N}{2}$ evaluation models on $D_{target}$, ensuring that each is initialized with a different seed. For the remaining $\frac{N}{2}$ evaluation models, we train on $D_{target}$ where $x_{target}$ is swapped out for a reference record $x_{ref}$, sampled from $D_{eval}$, again ensuring each has a different random seed. We estimate the privacy risk of $x_{target}$ by calculating a performance score over all $N$ evaluation models. The right-hand side of Figure 1 shows a diagram of this setup.

## 3.4 Comparison metrics

**RMSE.** We use Root Mean Squared Error ($RMSE$) to compare the results in the model-seeded and traditional evaluation setups. $RMSE$ quantifies the error made when using the traditional setup instead of model-seeded setup. Formally, we compute the error $\forall S \subseteq D_{target}$ as:

$$RMSE(S, \phi) = \sqrt{\frac{1}{|S|} \sum_{x \in S} (R_{trad}^{\phi} - R_{ms}^{\phi})^2}$$

Where $\bar{R}_{trad}^{\phi} = \frac{1}{|S|} \sum_{x \in S} R_{trad}^{\phi}$ is the mean traditional risk and $\bar{R}_{ms}^{\phi} = \frac{1}{|S|} \sum_{x \in S} R_{ms}^{\phi}$ is the mean model-seeded risk.

**Miss rate.** We define *miss rate* to be the fraction of records classified as high-risk in the model-seeded setup, that are classified as low-risk in the traditional setup. Given a threshold $t$, we consider a record $x$ to be high-risk if $R_{setup}^{\phi}(x, D) > t$, and low-risk otherwise, for $setup \in \{trad, ms\}$, dataset $D$, and metric $\phi$. Formally, we define the *miss rate* $\forall S \subseteq D_{target}$ as:

$$\mathcal{M}(S, D_{target}, D_{eval}) = \frac{|\{x \in S | R_{trad}^{\phi}(x, D_{eval}) \leq t \wedge R_{ms}^{\phi}(x, D_{target}) > t\}|}{|\{x \in S | R_{ms}^{\phi}(x, D_{target}) > t\}|}$$

Miss rate is dependent on high-risk threshold $t$, which is highly dependent on the setup. In our experiments, we select a value of $t$ that we consider reasonable, and report miss rate values for other values of $t$ in Appendix H.

## 4 A conceptual example

Previous work has shown outliers to be particularly susceptible to privacy attacks. However, outliers are dataset-specific, a record that is an outlier in one dataset may not be an outlier in another. Consequently, a record's privacy risk is likely to change depending on the dataset it is contained in. The traditional evaluation setup samples and averages across evaluation datasets where a target record may not consistently be an outlier. Thus, the target record's risk may be severely underestimated in this setup.

We illustrate the problem posed by the traditional evaluation setup using a toy example. We assume an attacker who aims to infer whether a target record, here a picture of a bird, was part of the training set of a model. By construction, we render the bird picture an outlier in the target dataset by sampling non-uniformly from CIFAR-10 (Krizhevsky et al., 2009). We then sample the evaluation dataset uniformly from CIFAR-10, making (artificially) the bird an outlier in the training dataset but likely not in the evaluation datasets. We then examine the risk reported by the both setups.

We denote $p = \frac{|\{x \in D_{target} | y(x) = \text{'bird'}\}|}{|D_{target}|}$ the fraction of the target dataset $D_{target}$ that we draw from the 'bird' class of CIFAR-10, where $y(x)$ denotes the class label of record $x$. The $(1 - p)|D_{target}|$ other images are drawn at random from the other 9 classes of CIFAR-10. As classes as equally represented, each will make up approximately $(1 - p)/9$ of the training dataset. We then compute the risk in the traditional setup and, for varying $p \in [0.0, 0.1]$, the model-seeded risk for our target 'bird' record in each dataset. To compute the traditional risk, we sample evaluation datasets from the evaluation pool in which all 10 classes are represented equally. We consider here target datasets of size 10,000.

Intuitively, if a bird were the only bird in the target dataset $D_{target}$, as is the case for low values of $p$, it would be a strong outlier with respect to $D_{target}$. Inferring whether this sample was seen by the model, compared to other records in $D_{target}$, would be easier for an attacker, thus increasing the record's vulnerability to MIAs (Carlini et al., 2022b; Meeus et al., 2023). Then, as $p$ increases, our 'bird' record is increasingly surrounded by other 'bird' records, becoming less of an outlier in $D_{target}$, decreasing its vulnerability to MIAs. However, as the traditional evaluation setup draws from an unbiased dataset, we show it to underestimate the risk for our target record, while our model-seeded setup provides a risk score realistic to the target dataset and the target record.

Figure 2a shows the traditional setup to estimate the risk to be 0.53, incorrectly evaluating the risk for specific target datasets, and most severely when the target record is an extreme outlier. Contrarily, the model-seeded setup correctly identifies our bird to be vulnerable, with AUC of 0.88, for very low values of $p$. It then identifies the risk to gradually decreases as more 'bird' records are included, approaching the risk score estimated by the traditional setup.

While artificial, this example shows how the traditional setup might incorrectly identify at risk records as it averages the risk across datasets.

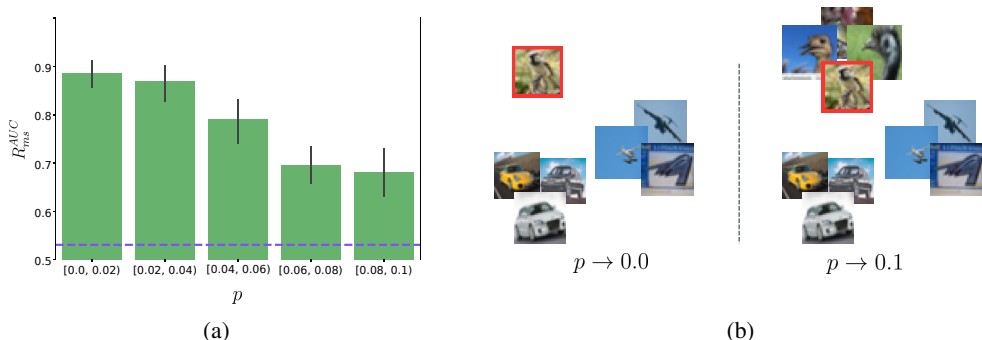

(a)                                                            (b)

Figure 2: (a) Model-seeded risk score of the target record plotted against the probability $p$ of sampling from the *bird* class. The lower $p$ is, the more of an outlier the target record is. (b) Visual interpretation of the effect of $p$ on the target dataset.

## 5 MIAS AGAINST GENERATIVE AND PREDICTIVE MODELS

**Experimental setup.** For synthetic data we use the SOTA MIA, the query-based attack introduced by Houssiau et al. (2022). We consider two models: synthetic data generators Synthpop (Nowok et al., 2016) and Baynet (Zhang et al., 2017), and two datasets: Adult (Becker & Kohavi, 1996) and UK Census (Office for National Statistics, 2011). $D_{eval}$ used to evaluate the MIA contains 15222 records for Adult and 27193 for UK census. We always consider $D_{target}$ to contain 1000 records. We compute the traditional and model-seeded privacy risk on all records in $D_{target}$ for Adult and, for computational efficiency, on a random subset of 100 records for UK census. The auxiliary dataset available to train the MIA contains 30000 records for Adult and 52390 records for UK census and is strictly not overlapping with $D_{eval}$. We train the MIAs using 1000 shadow models to ensure of the validity of our results (see Plot 5a and discussion in Appendix). Implementation details are given in Appendix I.

For ML classifiers, we consider two modalities: tabular and image classification. We train and evaluate three attacks from literature: LiRA (Carlini et al., 2022a), RMIA Zarifzadeh et al. (2024), and the modified prediction entropy attack (Mentr) proposed by Song & Mittal (2020). For image classification, we use a ResNet image classifier (He et al., 2016) trained on CIFAR10 (Krizhevsky et al., 2009) as our target model. We consider a target dataset $D_{target}$ of size $|D_{target}| = 10000$ and $|D_{eval}| = 30000$. For LiRA and Mentr, we use 256 evaluation models (as done by Carlini et al. (2022a)), and for RMIA, we use 16, as the authors state that a lower number of datasets is needed for this attack. For tabular data classification, the target model is a fully connected neural network classifier trained at predicting the binary salary attribute in Adult as the target models $\mathcal{M}_{target}$. We take $D_{target}$ of size $|D_{target}| = 2000$ and $|D_{eval}| = 15222$. We evaluate the effectiveness of LiRA and Mentr on 500 evaluation models, and RMIA on 25.

### 5.1 ADULT AND $\mathcal{M}_{target}$ SYNTHPOP

We start with a popular synthetic data generator, Synthpop, and the Adult dataset. In this section, we present the detailed results for this setup.

Figure 7a shows the AUC obtained by evaluating the MIA in the traditional setup to be an imperfect estimate of the effectiveness of the MIA against a model trained on the actual data used to train the

model to be released. Across all records, using the traditional setup leads to an RMSE of $0.07$, for a value that empirically ranges roughly from 0.5 to 1. Figure 7b shows that the risk score for a given target record would be off by more than 0.1 for $15\%$ of the records in the target dataset when using the traditional setup, and could go up to $0.26$. Figure 7a further shows that 94% of high-risk records would indeed be incorrectly classified as low-risk when estimating risk in the traditional setup.

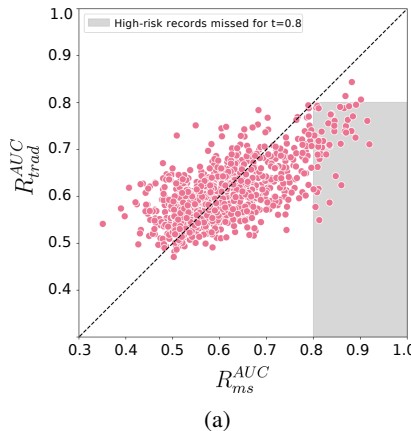
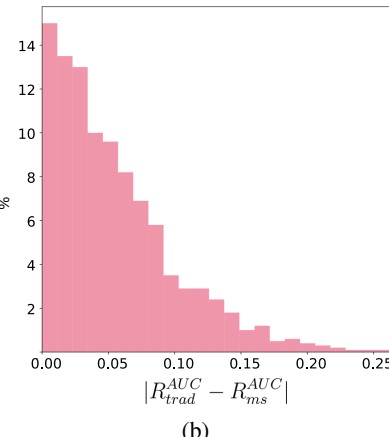

(a)          (b)

Figure 3: Privacy risk for each record in $D_{target}$ sampled from the Adult dataset, $\mathcal{M}_{target}$ Synthpop, and $\phi = AUC$. (a) per-record model-seeded and traditional risks. The dashed grey lines mark the high-risk threshold $t = 0.8$. The shaded grey area marks all the high-risk records missed in the traditional setup. (b) histogram of per-record absolute differences between the model-seeded and traditional risks.

## 5.2 EXTENDING TO OTHER GENERATORS, MODELS, AND DATASETS

The results we have presented so far were for one model and dataset. We now present the complete results across a total of 10 setups, varying across dataset, target model, and attack.

Table 1: Miss rate across different datasets, target models, and attacks. We use a high-risk threshold of $t = 0.8$. In the case where there are no records with a risk score higher than the threshold, the value is marked by '/'

| Dataset | $\mathcal{M}_{target}$ | Attack | $RMSE$ | Miss rate |
|---------|------------------------|--------|--------|-----------|
| Adult | Synthpop
Baynet | Query-based attack | 0.07
0.05 | 0.94
0.73 |
| Census | Synthpop
Baynet | Query-based attack | 0.11
0.04 | 0.94
0.75 |
| CIFAR10 | ResNet | LiRA
RMIA
Mentr | 0.07
0.14
0.10 | 0.40
0.52
0.29 |
| Adult | NN | LiRA
RMIA
Mentr | 0.05
0.11
0.16 | 1.00
1.0
/ |

Table 1 shows that we obtain similarly high RMSE across all setups, ranging from 0.05 to 0.16, a significant error for risk score estimated using AUC which typically ranges from 0.5 to 1.0. The miss rates show that this error is indeed causing incorrect identification of high-risk records. If the traditional setup were a good approximation of the risk we should observe miss rates close to 0. Instead, we observe values ranging from 30% to 94%, with the majority being above 50%. This means that, in most cases, more than half of the records that are highly vulnerable will be incorrectly

considered low-risk if evaluation is done in the traditional setup. We report miss rates across different high-risk thresholds in Table 4 in Appendix H. Interestingly, the miss rates for the ResNet image classifier trained on CIFAR10 are amongst the lowest we observe across datasets and models. We hypothesize this to be largely due to CIFAR10 being a much larger dataset ($|D_{target}| = 10,000$) than the other datasets considered ($|D_{target}| = 1000$), and outliers more often being preserved across large datasets.

# 6 Is $R_{trad}^{\phi}$ the mean of $R_{ms}^{\phi}$?

In our model-seeded setup, we eliminate what we argue is an incorrect source of randomness to evaluate MIAs against ML models: dataset sampling, as it does not provide a risk specific to the relevant dataset. We hypothesise that the risk calculated in the traditional setup is the mean of the risks specific to each dataset sampled in the traditional evaluation pipeline. In this section, we first provide empirical evidence for this hypothesis on a subset of the target dataset in one setup. The computational cost of empirically showing this hypothesis is very high, and it is infeasible to compute for a large number of records, or multiple setups–in the setup of Synthpop and Adult, the experiment takes 20 CPU hours for a single record, any would taken significantly more for the other setups. We therefore also present a theoretical result that is efficiently empirically validated, and provides additional support for our hypothesis.

## 6.1 Empirical validation

We first empirically validate our hypothesis for a random subset of 50 target records in our standard setup, i.e. Adult dataset and Synthpop generator.

For a given record, we first calculate the traditional risk $R_{trad}^{\phi}$ in the standard setup. Next, for 50 of the evaluation datasets sampled in the evaluation pipeline, we calculate the risk specific to the target record and each dataset, leading to 50 model-seeded risk scores. We then take the mean of the calculated model-seeded risks, which we denote $\tilde{R}_{sp}^{\phi}$. We calculate an RMSE of 0.02, and a correlation of 0.95 between the traditional risk $R_{trad}^{\phi}$ and $\tilde{R}_{sp}^{\phi}$, showing them to be very close in value. This provides us with empirical evidence for our hypothesis, and validates our claim that dataset sampling is an incorrect source of randomness when evaluating risk for a particular model or synthetic data generator release.

## 6.2 Theoretical support

To support our hypothesis, we present a theoretical result describing the relationship between two variables with a relationship consistent with our hypothesis. We first introduce and then prove (see Appendix B) the following theorem:

**Theorem 1.** *Let $X = [X_1, \cdots, X_n]$ where $\forall\, i \mid X_i$ is a random variable of mean $\mu_i$ and standard deviation $\sigma_i$, and let $\bar{X} = [\mu_1, \cdots, \mu_n]$. Then we have that the expected Pearson correlation between $X$ and $\bar{X}$ is equal to the square root of the ratio of their variances:*

$$E[\rho(X, \bar{X})] = \mathcal{V}(\bar{X}, X)$$

$$where\ \mathcal{V}(\bar{X}, X) = \sqrt{\frac{V(\bar{X})}{V(X)}}$$

With this, we then prove the relationship between two arrays $X$ and $\bar{X}$ of random variables with $\bar{X}_i$ being the mean of variable $X_i$.

Additionally, we note that

$$\bar{X} = E[X] \Rightarrow E[\bar{X}] = E[E[X]] = E[X]$$

i.e. the two arrays would have equal expected values.

In our context, $X$ would be the array of model-seeded risk estimates, while $\bar{X}$ would be the array of traditional risk estimates. If our hypothesis is correct, i.e. $R_{trad}^{\phi}$ is the average of $R_{ms}^{\phi}$, risk estimates

for records in $D_{target}$ would have a relationship consistent with Theorem 1. Subsequently, this would also empirically show that the randomness in the model-seeded setup associated with the sampling of the reference record only has a minor effect and is a good choice of randomness.

Empirically, this would mean that $\rho(S, \phi)$, the empirical correlation value, and $\mathcal{V}(R_{trad}^\phi, R_{ms}^\phi)$, the square root of variances value should be close to one another. Similarly, this means that $\bar{R}_{trad}^\phi$ and $\bar{R}_{ms}^\phi$ should be close to one another. To evaluate whether these relationships hold we compute $\rho(S, \phi)$ and $\mathcal{V}(R_{trad}^\phi, R_{ms}^\phi)$ in every setting of our experiments.

Table 2 shows how $\rho(S, \phi)$ and $\mathcal{V}(R_{trad}^\phi, R_{ms}^\phi)$ are close to each others across datasets and metrics and for both ML models and synthetic data generators. This empirical relationship between $R_{trad}^\phi$ and $R_{ms}^\phi$ is consistent with the hypothesis that the traditional risk is the mean of the model-seeded risk and provides evidence that model randomness is an appropriate source of randomness.

Table 2: $\rho(R_{trad}^\phi, R_{ms}^\phi)$ and $\mathcal{V}(R_{trad}^\phi, R_{ms}^\phi)$ values across different datasets, ML models and synthetic data generators. The attack used for ML models is LiRA.

|  | Synthpop | | Baynet | | Image clf. | Tab. clf. |
|---|---|---|---|---|---|---|
|  | Adult | Census | Adult | Census | CIFAR10 | Adult |
| $\rho$ | 0.65 | 0.85 | 0.68 | 0.86 | 0.88 | 0.79 |
| $\mathcal{V}$ | 0.63 | 0.85 | 0.75 | 0.83 | 0.81 | 0.91 |

## 7  MIA BY A STRONG ADVERSARY

We have so far compared evaluation setups using SOTA attacks from the literature. These assume the attacker to have access to an auxiliary dataset drawn from the same distribution but strictly non overlapping with $D_{target}$.

We now instantiate an MIA attack by a very strong attacker with knowledge of all the evaluation datasets including the training dataset $D_{target}$, similar to the standard DP attacker. In the traditional case, this means an attacker with access to the exact evaluation models–thus datasets sampled from $D_{eval}$–used for MIA evaluation. In the model-seeded case, this means an attacker with access to the full target dataset $D_{target}$.

We instantiate the stronger attack in our standard setup (Synthpop with the Adult dataset) and run the attack on 500 different target records randomly sampled from $D_{target}$. We compare the performance of the strong attacker to the attacker with only access to auxiliary data used previously (Sec. 5.1), which we refer to as the *classic* attack.

Fig. 4 shows the strong attacker to be able to achieve a substantially higher $AUC$ $0.792 \pm 0.092$ (mean and standard deviation) than the classic attacker with access to an auxiliary dataset ($0.601 \pm 0.092$) in the model-seeded setup. This is, however, not the case in the traditional setup where the strong attacker only achieves an AUC of $0.636 \pm 0.054$ compared to the $0.600 \pm 0.060$ achieved by the classic attacker.

These results further emphasize the need to use a model-seeded setup when evaluating MIAs against machine learning models, especially as new stronger attacks, leveraging knowledge of the training dataset, are likely to be developed in the future. Indeed, while we here focus on a very strong attacker, akin to the standard DP attacker, new attacks could leverage partial knowledge of the target dataset e.g. information leaked by the released synthetic dataset or knowledge about a specific part of the dataset where the target record would lie.

## 8  CONCLUSION AND FUTURE WORK

Our work shows that the current source of randomness used to evaluate MIAs against machine learning models is incorrectly averaging the risk across datasets. We instead propose to use a

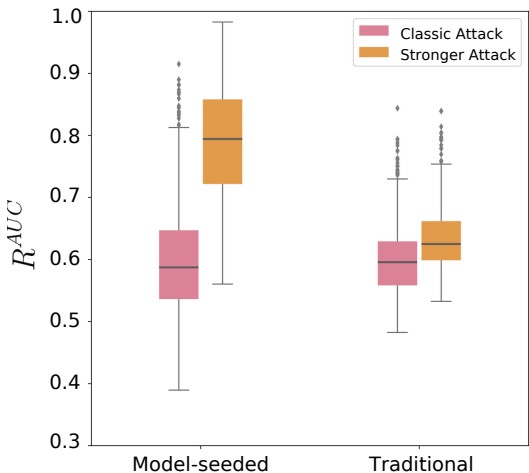

Figure 4: AUC results for the classic and stronger attack for 500 records randomly sampled from $D_{target}$, for both the traditional and the model-seeded setup.

model-seeded setup with model randomness as the only source of randomness. Using existing state-of-the-art MIAs, we compare the results obtained in the traditional setup with those obtained in the model-seeded setup and show them to lead to a high number of records to be misclassified as low risk. We show this to be true across 10 combinations of datasets, attacks, and ML models including generative and predictive models. We then derive theoretical results which, combined with empirical evidence, strongly suggest that the risk calculated in the traditional setup is an average of the risks specific to each dataset sampled for testing. We argue these results validate our use of model randomness as the (only) source of randomness necessary to evaluate MIAs against ML models. We finally instantiate an MIA by a very strong attacker and show the risk posed by this attack to be captured by the model-seeded setup while leading to the same result as the (much weaker) attacker with access to an auxiliary dataset in the traditional setup.

Taken together, our results strongly emphasize the need to evaluate the effectiveness of MIA attacks against machine learning model in the model-seeded setup. In particular, we show model randomness to be a good source of randomness.

Our work adds to the existing literature on privacy risk evaluation (Aerni et al., 2024; Stadler et al., 2022; Carlini et al., 2022b) and, in particular, enables more accurate record-level risk estimation when releasing machine learning models. We hope this work to help entities dealing with highly sensitive data, such as those in healthcare (Lotan et al., 2020) or the financial sector (Synthetic Data Expert Group, Financial Conduct Authority, 2024), to better understand the potential data leakage when releasing machine learning models and ensure a high standard of privacy. We believe this work to be particularly impactful for statistical "outliers", i.e. individuals whose data are significantly different from others, and to help ensure that their privacy is preserved to an equal standard as that of people whose data is closer to the average. Finally, we note that our results might lead to the development of stronger attacks which could be used by malicious adversaries. However, we believe that the knowledge our work provided to data controllers and for further research outweighs the potential risk.

## REPRODUCIBILITY

To ensure reproducible results, we provide detailed steps for both the traditional and model-seeded setups in Appendix D. We provide the libraries and datasets used for each experiment in Table 6 in Appendix I, all of which are publicly available. Table 7 provides the number of shadow and evaluation models used in each setup, as well as the size of the auxiliary dataset (used to train the attack), evaluation dataset, and target dataset. Finally, upon acceptance, we will release our codebase.

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

# A   APPENDIX

# B   SUPPLEMENTARY PROOF

**Theorem 1.** *Let $X = [X_1, \cdots, X_n]$ where $\forall\, i \mid X_i$ is a random variable of mean $\mu_i$ and standard deviation $\sigma_i$, and let $\bar{X} = [\mu_1, \cdots, \mu_n]$. Then we have that the expected Pearson correlation between $X$ and $\bar{X}$ is equal to the square root of the ratio of their variances:*

$$E[\rho(X, \bar{X})] = \mathcal{V}(\bar{X}, X)$$

$$\text{where } \mathcal{V}(\bar{X}, X) = \sqrt{\frac{V(\bar{X})}{V(X)}}$$

*Proof.* [Proof of Theorem 1] Let $X = [X_1, \cdots, X_n]$, with $\forall\, i \mid X_i$ being a random variable of mean $\mu_i$ and standard deviation $\sigma_i$, and let $\bar{X} = [\mu_1, \cdots, \mu_n]$.

Then, since $\rho(X, \bar{X}) = \frac{cov(X, \bar{X})}{\sqrt{V(X) \cdot V(\bar{X})}}$, by performing a member by member evaluation, we have:

$$\rho(X, \bar{X}) = \frac{\sum_i (X_i - \frac{1}{n}\sum_k X_k) \cdot (\mu_i - \frac{1}{n}\sum_k \mu_k)}{\sqrt{V(X) \cdot V(\bar{X})}}$$

which gives us :

$$E[\rho(X, \bar{X})] = \frac{\sum_i E[(X_i - \frac{1}{n}\sum_k X_k)](\mu_i - \frac{1}{n}\sum_k \mu_k)]}{\sqrt{V(X) \cdot V(\bar{X})}}$$

by linearity. Then, it follows

$$E[\rho(X, \bar{X})] = \frac{\sum_i (E[X_i] - \frac{1}{n}\sum_k E[X_k])(\mu_i - \frac{1}{n}\sum_k \mu_k)}{\sqrt{V(X) \cdot V(\bar{X})}}$$

$$= \frac{\sum_i (\mu_i - \frac{1}{n}\sum_k \mu_k)^2}{\sqrt{V(X) \cdot V(\bar{X})}}$$

$$= \frac{V(\bar{X})}{\sqrt{V(X) \cdot V(\bar{X})}}$$

$$E[\rho(X, \bar{X})] = \sqrt{\frac{V(\bar{X})}{V(X)}}$$

$\square$

## C AUC ROBUSTNESS

We run an experiment to determine the necessary number of evaluation models $N$ for the Adult dataset and $\mathcal{M}_{target}$ Synthpop. For value $N$, we perform the full traditional and model-seeded evaluation pipeline using $N$ evaluation models, and calculate and log the AUC. We repeat this 10 times for the same $N$, each time sampling new evaluation datasets in the traditional setup, and training new evaluation models in both setups. For each iteration, we calculate and log the AUC value. We then calculate the standard deviation of the 10 AUC values for $N$. We do this process for $N \in \{100, 200, 300, \cdots 2000\}$ for 10 target records. Figure 5a presents the standard deviations per $N$, aggregated over the 10 records. We select $N = 1000$ as the standard deviation converges to approximately 0.01 at that point.

We do the same experiment for Adult and $\mathcal{M}_{target}$ a binary classifier. In this case, we use 5 records and repeat the evaluation pipelines 5 times. We run for fewer records and rounds due to the high computational cost of this experiment. For $\mathcal{M}_{target}$ a binary classifier, we select $N = 500$, as it provides a sufficiently low standard deviation (0.02), while still allowing to feasibly conduct our main experiments.

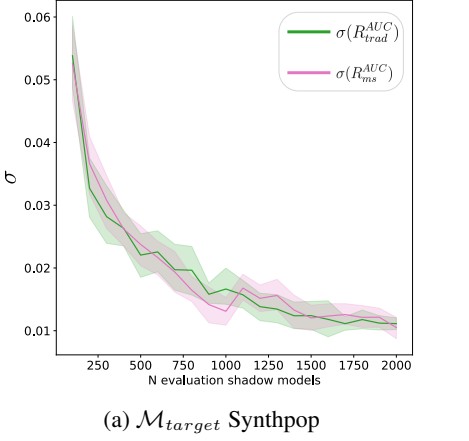
(a) $\mathcal{M}_{target}$ Synthpop

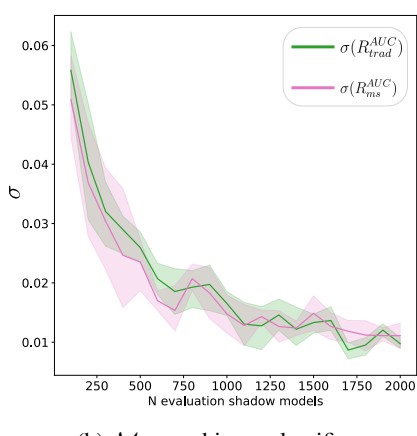
(b) $\mathcal{M}_{target}$ binary classifier

Figure 5: Standard deviation of the AUC as a function of the number of evaluation models used to calculate AUC Adult dataset.

## D DETAILED EVALUATION STEPS

In this section, we provide detailed steps for the traditional and model-seeded evaluation setups. Algorithm 1 presents the steps for the model-seeded evaluation setup. In Algorithm 2 we provide steps for evaluating an MIA in the traditional setup.

---

**Algorithm 1** Model-seeded MIA evaluation setup

---

Here we describe the pipeline for the model-seeded evaluation setup for membership inference attacks. We denote with $f_{\mathcal{M}}$ the trained meta-classifier (MIA). $D_{test}$ is the data available to the attacker which has not been used to train $f_{\mathcal{M}}$. $D_{target} \subset D_{eval}$ the target dataset, i.e. the dataset that will be used to train the model that will be released, $x_T \in D_{target}$ the target record, and reference record $x_R \in D_{eval}, x_R \notin D_{target}$.

1: Instantiate prediction set $Y_{pred} = \emptyset$.
2: Instantiate model initialization seed array $l_{seed}, |l_{seed}| = N$
3: **for** i $= 0 \cdots \frac{N}{2}$ **do**
4:     Train evaluation model $\mathcal{M}_{IN}$ with initialization seed $l_{seed}[2 * i]$
5:     Sample reference record $x_R$ from $D_{eval} \setminus D_{target}$
6:     Remove $x_T$ from $D$ and replace with $x_R$ to construct dataset $D_{OUT}$
7:     Train model $\mathcal{M}_{OUT}$ with seed $l_{seed}[2 * i + 1]$
8:     Add label-prediction pairs $(IN, f_{\mathcal{M}}(\mathcal{M}_{IN}))$ and $(OUT, f_{\mathcal{M}}(\mathcal{M}_{OUT}))$ to $Y_{pred}$
9: **end for**
10:     Calculate privacy risk using chosen metric based on prediction set $Y_{pred}$

---

**Algorithm 2** Traditional MIA evaluation setup

---

Here we describe the pipeline for the traditional evaluation setup for membership inference attacks. We denote with $f_{\mathcal{M}}$ the trained meta-classifier (MIA). $D_{eval}$ is the evaluation pool available to the attacker which has not been used to train $f_{\mathcal{M}}$. $D_{target} \subset D_{eval}$ is the target dataset, i.e. the dataset that will be used to train the model that will be released, and $x_T \in D_{target}$ is the target record.

1: Instantiate prediction set $Y_{pred} = \emptyset$.
2: Instantiate the training seed array $l_{seed}, |l_{seed}| = N$
3: **for** i $= 0 \cdots \frac{N}{2}$ **do**
4:     Sample $D_{IN} \sim D_{eval} \setminus \{x_T\}, |D_{IN}| = |D_{target}| - 1$, and add $x_T$ to $D_{IN}$
5:     Sample $D_{OUT} \sim D_{eval} \setminus \{x_T\}, |D_{OUT}| = |D_{target}|$
6:     Train evaluation models $\mathcal{M}_{IN}$ with seed $l_{seed}[2 * i]$ and $\mathcal{M}_{OUT}$ with $l_{seed}[2 * i + 1]$).
7:     Add label-prediction pairs $(IN, f_{\mathcal{M}}(\mathcal{M}_{IN}))$ and $(OUT, f_{\mathcal{M}}(\mathcal{M}_{OUT}))$ to $Y_{pred}$
8: **end for**
9: Calculate privacy risk using chosen metric based on prediction set $Y_{pred}$

---

## E VISUALISATION OF RESULTS FOR CIFAR-10

We show in Figure 6 the model-seeded and traditional risks for CIFAR-10.

## F OUTLIER MEASURE OF MISCLASSIFIED RECORDS

To gain insight into the level to which the misclassified records are outliers in the target dataset, we perform the following analysis: For each record in the target dataset, we calculate an "outlier measure" as defined by Meeus et al. (2023). This measure is determined by the mean distance of a record to its 100 nearest neighbors. For CIFAR10, we computed these distances using the embeddings derived from the output of the last linear layer of a ResNet model trained on the full CIFAR10 training dataset, which consists of 30,000 records. Figure 7 shows the distribution of the outlier measures of the records in the target dataset, and the distribution of the outlier measures of the misclassified records. The results show that the misclassified records tend to have higher outlier scores. However, some misclassified records are closer to the average, indicating that while outliers do tend to be particularly vulnerable, being an outlier is not the sole indicator of misclassification, and there are other factors that affect a record's risk.

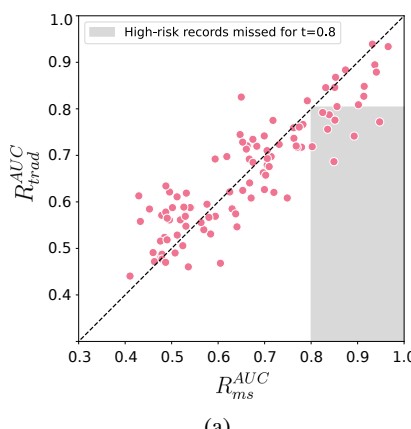
(a)

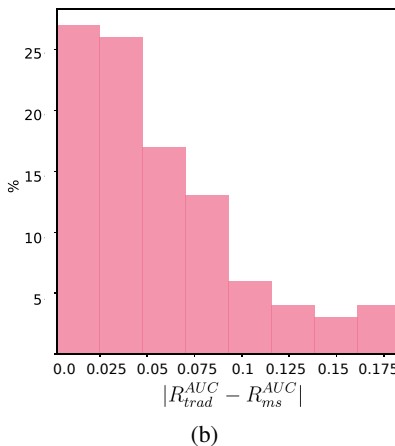
(b)

Figure 6: Privacy risk for 100 records in $D_{target}$ sampled from the CIFAR-10 dataset, $\mathcal{M}_{target}$ ResNet, and $\phi = AUC$. (a) per-record model-seeded and traditional risks. The dashed grey lines mark the high-risk threshold $t = 0.8$. The shaded grey area marks all the high-risk records missed in the traditional setup. (b) histogram of per-record absolute differences between the model-seeded and traditional risks.

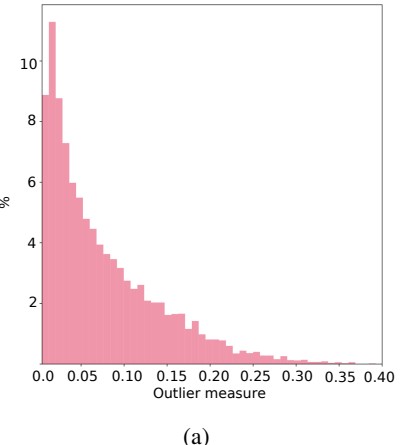
(a)

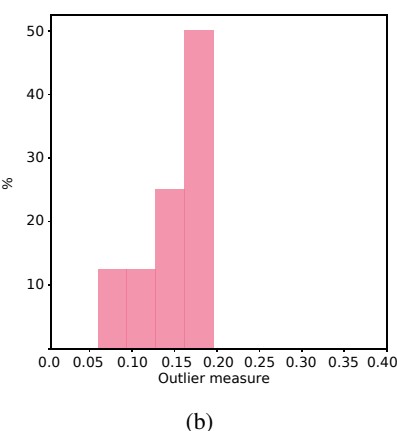
(b)

Figure 7: Distribution of outlier measures of records in the target dataset for CIFAR-10. (a) histogram of outlier measures of all records in the target dataset. (b) histogram of outlier measures in the target dataset incorrectly identified as low-risk in the traditional evaluation setup.

## G RESULTS USING ACCURACY AND DISCUSSION ON EVALUATION METRICS

Table 3 shows the RMSE values across all setups when accuracy is used to calculate the risk score. In this work, we use ROC AUC and accuracy as performance metrics for MIAs. Here we note that though TPR at low FPR is currently the most widely-used metric for evaluating MIAs (Carlini et al., 2022a; Zarifzadeh et al., 2024; Song & Mittal, 2020), the information it provides is not relevant to evaluating per-record attacks. TPR at low FPR is specifically used when applying a general attack to a set of records, helping to identify records for which the attack confidently determines membership. As we train and evaluate a separate attack for each record, TPR at low FPR would not be informative in the same way.

## H MISS RATE FOR DIFFERENT HIGH-RISK THRESHOLDS

Table 4 shows the miss rate for the counting query attack on synthetic data generators Synthpop and Baynet, for the Adult and UK Census datasets.

Table 3: $RMSE$ across different datasets and synthetic data generators for performance measured with accuracy.

| Dataset | $\mathcal{M}_{target}$ | Attack | $RMSE$ |
|---------|------------------------|--------|--------|
| Adult | Synthpop
Baynet | Query-based attack | 0.05
0.04 |
| Census | Synthpop
Baynet | Query-based attack | 0.09
0.03 |
| CIFAR10 | ResNet | LiRA
RMIA
Mentr | 0.12
0.12
0.08 |
| Adult | NN | LiRA
RMIA
Mentr | 0.04
0.06
0.1 |

Table 4: Miss rate for high-risk thresholds between $0.5$ and $0.9$ for the counting query attack on synthetic data generators. In case there are no records with a risk score higher than the threshold, the value is marked by '/'

| Dataset | $\mathcal{M}_{target}$ | $t$ | Miss rate | |
|---------|------------------------|-----|-----------|----------|
| | | | AUC | accuracy |
| Adult | Synthpop | 0.5 | 0.01 | 0.01 |
| | | 0.6 | 0.27 | 0.43 |
| | | 0.7 | 0.60 | 0.71 |
| | | 0.8 | 0.94 | 1.0 |
| | | 0.9 | 1.0 | / |
| | Baynet | 0.5 | 0.09 | 0.10 |
| | | 0.6 | 0.29 | 0.21 |
| | | 0.7 | 0.27 | 0.25 |
| | | 0.8 | 0.73 | 0.84 |
| | | 0.9 | 0.86 | / |
| UK Census | Synthpop | 0.5 | 0.01 | 0.01 |
| | | 0.6 | 0.27 | 0.43 |
| | | 0.7 | 0.60 | 0.71 |
| | | 0.8 | 0.94 | 1.0 |
| | | 0.9 | 1.0 | / |
| | Baynet | 0.5 | 0.15 | 0.14 |
| | | 0.6 | 0.60 | 0.50 |
| | | 0.7 | 0.78 | 0.80 |
| | | 0.8 | 0.75 | / |
| | | 0.9 | / | / |

Table 5: Miss rate for high-risk thresholds between $0.5$ and $0.9$ for the LiRA on synthetic data generators. In case there are no records with a risk score higher than the threshold, the value is markes by '/'

| Dataset | $\mathcal{M}_{target}$ | $t$ | LiRA | | RMIA | | Mentr | |
|---|---|---|---|---|---|---|---|---|
| | | | Miss rate | | Miss rate | | Miss rate | |
| | | | AUC | accuracy | AUC | accuracy | AUC | accuracy |
| CIFAR-10 | ResNet | 0.5 | 0.04 | 0.04 | 0.05 | 0.05 | 0.04 | 0.11 |
| | | 0.6 | 0.07 | 0.08 | 0.18 | 0.24 | 0.05 | 0.30 |
| | | 0.7 | 0.21 | 0.44 | 0.35 | 0.50 | 0.29 | 0.80 |
| | | 0.8 | 0.40 | 0.78 | 0.52 | 0.79 | 0.30 | 1.0 |
| | | 0.9 | 0.75 | 1.0 | 0.80 | 1.0 | 1.0 | / |
| Adult | NN | 0.5 | 0.19 | 0.10 | 0.30 | 0.20 | 0.78 | 0.59 |
| | | 0.6 | 0.36 | 0.71 | 0.66 | 0.73 | / | / |
| | | 0.7 | 0.71 | 0.88 | 0.80 | / | / | / |
| | | 0.8 | 1.0 | 1.0 | 1.0 | / | / | / |
| | | 0.9 | 1.0 | 1.0 | / | / | / | / |

Table 6: Datasets and libraries used for each experiment setting.

| $\mathcal{M}_{target}$ | Datasets | Library |
|---|---|---|
| Synthpop (Nowok et al., 2016) | Adult (Becker & Kohavi, 1996) | Reprosyn |
| Baynet (Zhang et al., 2017) | UK Census (Office for National Statistics, 2011) | (Alan Turing Institute, 2022) |
| ResNet (He et al., 2016) | CIFAR-10 (Krizhevsky et al., 2009) | PyTorch (Paszke et al., 2019) FFCV (Leclerc et al., 2023) |
| Neural Network | Adult | scikit-learn Pedregosa et al. (2011) |

Table 5 shows the miss rate for the LiRA, RMIA and Mentr attacks on ML classifiers. For CIFAR-10 the target model is ResNet, and for Adult it is a fully connected neural network.

## I Experimental Setup

Table 6 shows the libraries used for implementing each target model and attack, as well as the datasets used for each setting. Table 7 shows the number and size of evaluation and shadow datasets, the size of the target dataset, and the size of the reference dataset used for each attack. We train a standard ResNet and a neural network with 1 layer with 100 nodes.

Table 7: Experiment details for each attack. $N_z$ refers to the number of reference records used for the RMIA attack.

Table 7: Experiment details for each attack. $N_z$ refers to the number of reference records used for the RMIA attack.

| Dataset | Attack | $N_{shadow}$ | $N_{eval}$ | $N_z$ | $|D_{aux}|$ | $|D_{eval}|$ | $|D_{target}|$ |
|---|---|---|---|---|---|---|---|
| Adult | Counting query attack | 1000 | 1000 | / | 30000 | 15222 | 1000 |
| UK Census | (Houssiau et al., 2022) | 1000 | 1000 | / | 52390 | 27193 | 1000 |
| CIFAR-10 | LiRA (Carlini et al., 2022a) | 256 | 256 | / | 30000 | 30000 | 10000 |
| | Mentr Song & Mittal (2020) | 256 | 256 | / | 30000 | 30000 | 10000 |
| | RMIA Zarifzadeh et al. (2024) | 16 | 16 | 2500 | 30000 | 30000 | 10000 |
| Adult | LiRA | 500 | 500 | / | 30000 | 15222 | 2000 |
| | Mentr | 500 | 500 | / | 30000 | 15222 | 2000 |
| | RMIA | 25 | 25 | 500 | 30000 | 15222 | 2000 |

