# OpenReview forum: "Lost in the Averages: Evaluating record-specific MIAs against Machine Learning models"
_ICLR.cc/2025/Conference — Submitted to ICLR 2025_

### Official Review · Reviewer_Z8YX · 2024-11-02

**Soundness:** 2
**Presentation:** 2
**Contribution:** 2
**Rating:** 5
**Confidence:** 4

**Summary:**

The paper introduces a new evaluation setup, termed model-seeded evaluation, to assess privacy risks for individual records in Membership Inference Attacks (MIAs) against machine learning models. The traditional MIA evaluation setup relies on dataset sampling as a randomness source, which can obscure risks for outlier records, as these outliers may not consistently appear across sampled datasets. In contrast, the model-seeded approach keeps the dataset fixed and varies the model’s randomness (e.g., initialization seeds and training variations) to better capture privacy risks, especially for outliers. The authors validate this approach through experiments with various models, datasets, and attacks, showing that model-seeded evaluation more accurately assesses per-record privacy risks.

**Strengths:**

1.Clarity: The paper is well-organized.

2.Originality: The proposal of model-seeded evaluation as a way to focus on model randomness rather than dataset sampling introduces a new angle for evaluating privacy risks in MIAs.

**Weaknesses:**

1.Lack of Clear Novelty and Contribution: It is challenging to grasp the novelty and practical contribution of the proposed method within the scope of the ICLR main conference. This work might be more suitable for another conference focusing on evaluation methods rather than core ML contributions or a Datasets and Benchmarks Track.

2.Confusion Regarding Proposal: There is ambiguity in the proposal's application, especially regarding the handling of shadow models. For example, in "Membership Inference Attacks from First Principles" (S&P 2022), shadow models rely on training diversity achieved through different sampling. If the proposal implies keeping the majority of training data fixed and varying model seeds, this approach may fail to capture the diversity that shadow models achieve through varying data, potentially resulting in inaccurate representations of privacy risks.

**Questions:**

I am confused by the implementation of your proposal. I would like the authors to address the following specific question:
In "Membership Inference Attacks from First Principles" (S&P 2022), shadow models are trained with IN-models and OUT-models that derive their diversity primarily from different training samples. Is your proposal to maintain most of the training data unchanged and instead vary only the model seeds and initialization weights to achieve diversity among shadow models? If so, this method might be flawed, as it could produce shadow models that do not reflect the target model's distribution accurately, especially if there are significant differences between the fixed training data for shadow models and the target model’s data.
To illustrate, Figure 3 in "Membership Inference Attacks from First Principles" shows that the variability in IN models with different training samples significantly impacts privacy risk assessments. Using fixed training data with only model randomness may lead to substantial errors in evaluating privacy risk, as it could miss the model diversity required to mimic target model behavior. Could the authors clarify this approach? If my understanding is incorrect, please correct it. Thank you.

Decision: I am giving a score of 5 (marginally below the acceptance threshold). While I appreciate the attempt at a new evaluation method, if my understanding is correct, this proposal may be inherently flawed. I look forward to the authors' responses to my concerns.

---

> ### Author Response · Authors · 2024-11-20
> **Response to raised questions and weaknesses**
>
> We thank the reviewer for their thoughtful response.
>
> We would like to clarify the distinction between MIA development (e.g. training a meta-classifier, or developing a scoring function based on hypothesis testing) and evaluation (evaluating how the developed MIA performs at inferring membership in unseen settings). For both, IN and OUT models are needed.
>
> We believe the reviewer refers to IN and OUT models used for MIA development. For this, we apply exactly the same setup as in “Membership Inference Attacks from First Principles”, training IN and OUT shadow models by sampling datasets from an auxiliary dataset which stems from the same distribution as the target dataset (but non-overlapping).
>
> In contrast, our contribution focuses on MIA evaluation. We instantiate first the traditional evaluation setup, which is a natural extension of the shadow modeling setup used for MIA development (sampling datasets from a larger test dataset, non-overlapping with the auxiliary dataset). We argue that this setup does not allow for the accurate computation of record-level risk for a record as part of a specific target dataset, as it averages the risk across many randomly sampled datasets. To measure this appropriately, we propose the model-seeded evaluation setup, where the evaluation shadow datasets always closely correspond to the target dataset while creating the randomness across evaluation models using training randomness.
>
> Our results show that the risk of a record is indeed influenced by the other records in the dataset, and that averaging the risk (as in the traditional setup) can cause high-risk records to be incorrectly identified as low-risk, and vice-versa. Because of this, we believe it is essential to evaluate MIAs in the model-seeded setup when evaluating the privacy risk of a machine learning release.

---

> ### Comment · Reviewer_Z8YX · 2024-11-27
> **Official Comment by Reviewer Z8YX**
>
> Thank you for your response. I understand the context you're working with. However, as a researcher in MIA, I believe that major researchers in this field would not consider your assumption that the attacker has access to the target model's training dataset as reasonable. This assumption contradicts the fundamental premise of MIAs. None of the previous MIAs [1][2][3], operate under this assumption.
>
> I would recommend referring to the paper Membership Inference Attacks against Machine Learning Models, which is the first paper in the MIA field. The reason shadow models are used in MIA is precisely because the attacker cannot access the target model's training data. If the attacker had access to the training data, the methods outlined in this paper would not need to be so complicated; a simple classifier could be trained directly on the training data versus non-training data.
>
> Regarding the method presented in this submission, it is not an improvement on Membership Inference Attacks from First Principles, nor does it align with the views of major researchers in the MIA field, who all agree that the attacker should not have access to the target model's training set. The modifications proposed in the submission are quite limited, and it is well understood by MIA researchers that if the attacker had access to the training data, attacker performance would naturally be much higher.
>
> I believe the authors should have conducted a more thorough background review of the MIA field before presenting this work. I do not consider this submission to be a reasonable paper, though it could be a technical report. I maintain my score. Once again, thank you for your response and good luck for your future work.
>
> [1] Zarifzadeh, et al. Low-Cost High-Power Membership Inference Attacks. ICML 2024.
>
> [2] Carlini, N., et al. Membership inference attacks from first principles. S&P 2022.
>
> [3] Ye, J., et al. Enhanced membership inference attacks against machine learning models. CCS 2022.
>
> [4] Membership Inference Attacks against Machine Learning Models. In SP 2017.

---

### Official Review · Reviewer_1E7s · 2024-11-03

**Soundness:** 1
**Presentation:** 2
**Contribution:** 2
**Rating:** 3
**Confidence:** 4

**Summary:**

The paper challenges the traditional method for evaluating record-specific MIA on machine learning models, which assesses privacy risks by averaging results across various datasets. This conventional method, which uses dataset sampling as a source of randomness, may not accurately capture the risks associated with outlier records. These records, which are statistically distinct, often carry higher privacy risks. To address this, the authors propose a "model-seeded" evaluation, relying on model-specific randomness rather than dataset variation. This approach is argued to be more sensitive to record-level risk, especially for outliers, thus providing a more accurate measurement of privacy threats posed by MIA.

**Strengths:**

The paper challenges the traditional method for evaluating record-specific MIA on machine learning models, which assesses privacy risks by averaging results across various datasets. This conventional method, which uses dataset sampling as a source of randomness, may not accurately capture the risks associated with outlier records. These records, which are statistically distinct, often carry higher privacy risks. To address this, the authors propose a "model-seeded" evaluation, relying on model-specific randomness rather than dataset variation. This approach is argued to be more sensitive to record-level risk, especially for outliers, thus providing a more accurate measurement of privacy threats posed by MIA.

**Weaknesses:**

1.The model-seeded method assumes that outlier status consistently correlates with higher privacy risks, a premise that might not apply universally across all datasets or model types. This limitation is also notable, as generalizing outliers as inherently high-risk without broader evidence may lead to misjudgments. If outliers do not uniformly correlate with increased privacy risks, adopting this assumption could skew privacy evaluations and limit the model-seeded approach’s applicability.

2.By focusing on model-seeded evaluations, this method may potentially amplify perceived privacy risks for records that happen to exhibit outlier characteristics, which might lead to overly conservative evaluations. Though less immediately impactful, an overestimation of privacy risks could have negative implications, potentially stifling the release of data and models in scenarios where the actual privacy risk might be minimal.

3.The most critical limitation, as the added computational requirements could limit the adoption of the model-seeded approach in practice. Evaluating privacy risks in real-world scenarios often demands rapid processing across various models and datasets, and an overly resource-intensive method could hinder its scalability and accessibility.

**Questions:**

1.The author critiques the lack of true randomness in traditional sampling-based evaluation metrics, arguing that these metrics fail to accurately capture privacy risks in models. However, non-randomness in sampling does not necessarily lead to bias in evaluation outcomes. The issue of outliers disproportionately affecting privacy risk assessments arises primarily when the entire training dataset is consistently used across each model being attacked. By performing adequate and independent random sampling of subsets, the aggregate distribution of sampled datasets would approximate that of the full training set, reducing outlier impact. This raises a question about the necessity of model-seeded sampling in place of random sampling. When each model is trained on the complete dataset, outliers indeed amplify the privacy risk assessment. However, ensuring random sampling alone does not necessarily align the distribution between training and non-training sets for evaluation, which can potentially affect the fairness of the evaluation metrics.

2.In the Related Work section, the author overlooks key recent studies specifically addressing advancements in MIA evaluation metrics. Expanding this review to include recent literature on MIA evaluation techniques would better contextualize the current study within the broader research landscape. Additionally, the statement beginning "The standard evaluation setup for MIAs..." on Line 124, Page 3 lacks clarity on why conventional metrics fall short or what specific limitations they exhibit in this context. Providing concrete examples or detailed limitations of these standard metrics would clarify the rationale for proposing alternative methods. Comparative examples would enhance readers’ comprehension of the need for innovative evaluation approaches.

---

> ### Author Response · Authors · 2024-11-20
> **Responses to weaknesses and questions**
>
> We thank the reviewer for their time and their insightful comments on our work. Below are answers to each weakness and question individually.
>
> **Outliers (W1,2; Q1):**
>
> Our work argues that using the model-seeded setup provides a better estimate for a real-life model release. Indeed, while only one model trained on one (target) dataset will be released, the average setup, using data sampling, averages the risk across many randomly sampled datasets. We do not assume that being an outlier is a perfect indicator of high risk, but previous work (Carlini et al, Meeus et al) has shown that outliers do tend to be more vulnerable to attacks. Outliers are, however, not necessarily consistent across different random samples (such as the datasets randomly sampled in the traditional evaluation setup). This indicates that the risk of a record may also vary across different samples, and that estimating the risk by averaging across random samples could lead to an incorrect estimate.
>
> Figure 7 in Appendix F in the revised (current) version of our paper shows the distribution of the outlier measures of the records in the target dataset, and the distribution of the outlier measures of the misclassified records. The results show that the misclassified records tend to have higher outlier scores, while some misclassified records are closer to the average. This indicates that while outliers do tend to be particularly vulnerable, being an outlier is not the sole indicator of misclassification, and there are other factors that affect a record’s risk.
>
> **Computational resources (W3):**
>
> The model-seeded setup is not more computationally expensive than the traditional setup - both setups require training the equal number of evaluation models, where the only difference is the datasets that the evaluation models are trained on. While record-level MIAs can be more computationally expensive than dataset-level MIAs, they are essential for accurate risk estimation for specific records (Meeus et al, Long et al, Stadler et al). Both the traditional and model-seeded setups are record-specific.
>
> **Related work (Q2):**
>
> Our work focuses specifically on record-level risk evaluation such as, for example, by Stadler et al. and Meeus et al. While recent work, such as work by Aerni et al, has identified critical issues in MIA evaluation and proposed improvements, we identify a specific issue in record-level risk estimation - averaging the risk estimate across randomly sampled datasets. We argue that averaging across these datasets leads to an incorrect estimate, which we show to be true in our results.
>
> References:
>
> *Nicholas Carlini, Matthew Jagielski, Chiyuan Zhang, Nicolas Papernot, Andreas Terzis, and Florian
> Tramer. The privacy onion effect: Memorization is relative. Advances in Neural Information
> Processing Systems, 35:13263–13276, 2022b.*
>
> *Matthieu Meeus, Florent Guepin, Ana-Maria Cretu, and Yves-Alexandre de Montjoye. Achilles’ heels: vulnerable record identification in synthetic data publishing. In European Symposium on Research in Computer Security, pp. 380–399. Springer, 2023*
>
> *Yunhui Long, Lei Wang, Diyue Bu, Vincent Bindschaedler, Xiaofeng Wang, Haixu Tang, Carl A
> Gunter, and Kai Chen. A pragmatic approach to membership inferences on machine learning
> models. In 2020 IEEE European Symposium on Security and Privacy (EuroS&P), pp. 521–534.
> IEEE, 2020.*
>
> *Theresa Stadler, Bristena Oprisanu, and Carmela Troncoso. Synthetic data – anonymisation groundhog day, 2022.*

---

### Official Review · Reviewer_kAVu · 2024-11-03

**Soundness:** 1
**Presentation:** 2
**Contribution:** 1
**Rating:** 3
**Confidence:** 4

**Summary:**

The paper introduces a new framework for quantifying privacy risks through membership inference attacks. The authors argued that the traditional approach assesses the effectiveness of these attacks by comparing models trained with the target data point against models trained without it, with the remaining training data randomly sampled from the data pool. In contrast, this paper proposes a refined setup: the remainder of the dataset is held constant, and the performance of the membership inference attack is evaluated based on its ability to distinguish between models trained with and without the target point while keeping all other training data identical.

**Strengths:**

The paper studies privacy auditing using membership inference attacks, which is an important and interesting topic.

**Weaknesses:**

1. The paper seems unclear about why different evaluation setups are needed. In my understanding, membership inference attacks can be framed as a game between an adversary and the model trainer, where the privacy risk is quantified by the adversary's success in this game (Ye et al. 2022, Sablayrolles et al. [1], Yeom et al. [2], Carlini et al. (2022a)). Each setup defines the privacy risk in a distinct context, such as the average privacy risk of a training algorithm or the specific privacy risk associated with a single data point. To quantitatively assess privacy risks, different setups are designed to simulate these privacy games. Traditional setups and model-seeded setups represent two distinct privacy games and, as a result, inherently measure different types of privacy risks. Therefore, comparing metrics such as RMSE and Miss Rate across these two setups is not meaningful.

2. The new proposed setups are identical to the Definition 3.4 of Section 3.4 in Ye et al. 2022 and the paper lacks discussion in this work.

3. The paper uses ROC AUC and accuracy to assess privacy risks, which is problematic, as discussed extensively in Carlini et al. (2022a). Instead, the evaluation should include at least the true positive rate at a low false positive rate.

[1] Alexandre Sablayrolles, Matthijs Douze, Cordelia Schmid, Yann Ollivier, and Hervé Jégou. 2019. White-box vs black-box: Bayes optimal strategies for membership inference. In International Conference on Machine Learning.

[2] Samuel Yeom, Irene Giacomelli, Matt Fredrikson, and Somesh Jha. 2018. Privacy risk in machine learning: Analyzing the connection to overfitting. In 2018 IEEE 31st Computer Security Foundations Symposium (CSF).

**Questions:**

1. What is the privacy game that this paper investigates, and how are the new setups designed to simulate it? (see w1). To address these comments, I would like to ask the authors to explicitly define the privacy game for the MIA settings discussed in the paper and justify why comparing metrics across setups is meaningful in this context.

2. Please compare with the evaluation setups in Definition 3.4 in Section 3.4 of Ye et al. 2022 and justify what is new compared to the existing evaluation setups. In particular, in Definition 3.4 of Ye et al. (2022), the adversary’s objective is to differentiate between models trained on fixed subsets that include the target point and those trained on the fixed subsets without the target point. Similarly, the proposed setups, as illustrated in Figure 1, aim to distinguish between a model trained on $D_{\text{target}}$ and one trained on $ D_{\text{target}}$ with $x_{\text{target}}$ replaced by $x_r$.

3. What is the relationship between the target model and the evaluation model? Specifically, are you inferring membership for a fixed target model (which relates to the privacy risk of a specific trained model), or are you using a set of evaluation models to quantify the privacy risk for a single target sample?

4. Please also report the empirical results (Table 1) and the theoretical analysis (Theorem 1) based on the true positive rate at the low false positive rate (see w3).

---

> ### Author Response · Authors · 2024-11-20
> **Responses to weaknesses and questions**
>
> We thank the reviewer for their insightful and thorough comments on our paper. We answer each of the raised weaknesses and questions individually below.
>
> **Privacy game:**
>
> The two evaluation setups do indeed represent two distinct privacy games, which define the evaluation of the MIA, but do not, in the context of our main experiments, affect the training of the MIA. For a given target record we train one MIA, and then evaluate it in both setups.
>
> Our work indeed precisely shows that record-level privacy risk is highly influenced by the dataset the record is in. Recent work has shown that outliers tend to be highly vulnerable. Since outliers may not consistently appear across different data samples, we hypothesise that high-risk records may similarly be inconsistent across samples. As a result, evaluating in the traditional setup may result in records being incorrectly identified as low-risk, and, more generally, incorrect risk estimation, regardless of the auxiliary knowledge of the attacker or the training procedure of the MIA.
>
> **Comparison with Ye et al:**
>
> Our work focuses on record-level risk evaluation, independent of the training of the MIA itself. Definition 3.4 in Ye et al., on the other hand, describes a worst-case attacker - the MIA is trained with knowledge of the full target dataset except for the target record.
>
> We argue that the model-seeded setup should be used for evaluation regardless of the attacker’s knowledge. We here implement the standard adversary - we train an MIA with access to an auxiliary dataset drawn from the same distribution as the target dataset, but with no knowledge of the target dataset, and show the traditional and model-seeded risks of the same MIA to be different. Evaluating the MIA in the model-seeded setup enables the data owner to more closely estimate the risk related to the target record and the actual dataset which will be used to train the model that will be released.
>
> **Relationship between the target model and the evaluation model:**
>
> Our goal is to estimate the risk of a target record in a specific (target) dataset. We put ourselves in the role of the data owner, wanting to release a model trained on the target dataset, and we aim to estimate the risk of a specific record in the target dataset. The evaluation models are trained on the target dataset, and the risk estimate is calculated solely based on the MIA’s performance on these evaluation models, allowing us to more closely estimate the success of an MIA against the model that will be trained on the target dataset and released.
>
> **TPR at low FPR:**
> In “Membership Inference Attacks from First Principles”, the authors evaluate the MIA across all records in the dataset, while we focus on per-record evaluation. TPR at low FPR acts as a dataset-level metric: when evaluating across multiple records, it allows the data owner to determine the records for which the MIA confidently and correctly infers membership. This makes it particularly relevant for understanding the overall risk of a dataset.
>
> However, in our context, where we train and evaluate an MIA separately for each record, TPR at low FPR reflects the specific seeds used to train the evaluation models where the MIA performs well. As the seed is random and not a property of the dataset nor the record, we do not consider this particularly meaningful in this context.

---

### Official Review · Reviewer_x1tv · 2024-11-03

**Soundness:** 3
**Presentation:** 4
**Contribution:** 3
**Rating:** 6
**Confidence:** 4

**Summary:**

This paper argues that recently proposed MIAs donot correctly capture the privacy risks of training samples. The main argument is that the high risk samples are often outliers, and outliers are dependent to the sampled datasets. Thus shadow models trained with randomly sampled datasets may not be able the reflect the fact that the underlying sample is an outlier in the trainingset of target model.

To address this, the authors propose to obtain randomness by vary different weight initialization and training seeds.  The experimental results show that the proposed model-seed based approach can detect many high risk samples which are missed by traditional MIAs such as LiRA.  Moreover, when dealing with the strongest DP adversary, the proposed model-seed approach also give higher attack success rate compared with traditional ones.

In addition, the authors also argue that TPR at low FPR metric is not relevant in evaluating per-record MIAs, thus they opt to use ROC AUC as the privacy metric.

**Strengths:**

1. The idea of randomizing model seeds is novel. Also the observation that randomizing training datasets in shadow models might not be able to capture whether a specific sample is an outlier in the trainingset of the target model or not, is reasonable.

2. The paper gives a very nice and clear overview of MIAs, I appreciate it.

3. The two newly defined metrics demonstrate the improvement of model-seeded compared to traditional methods.

4. Comprehensive cross-method experiments.

**Weaknesses:**

1. The privacy risk calculation process is not clear enough.

2. See questions below

**Questions:**

1. In Figure 2a, where is the value 0.53 indicated? Is it represented by the dashed line?

2. It is unclear how exactly the privacy risk is calculated. In Algorithm 1, it states to "calculate privacy risk using a chosen metric based on the prediction set." How is this done in practice? For each sample, do you determine the best threshold that separates IN and OUT models and then use this threshold to compute the AUC?

3. The argument that TPR at low FPR is unsuitable for assessing per-record MIAs is not fully explained. Why is the TPR at low FPR, as proposed in LiRA, not appropriate for evaluating the privacy of per-record MIA?

4. The distinction between evaluating model privacy and record-level privacy needs clarification. If my understanding is correct, the authors suggest that existing MIAs generally measure how a particular target model reveals its training data's membership information, rather than focusing on the privacy risk of a specific sample when included in the training set. While I agree with the motivation for record-specific MIAs, these two cases are closely related. Whether a sample is an outlier depends not only on the data distribution but also on the deployed models. The strictest setting, following differential privacy (DP), assumes full knowledge of the dataset except for the sample being evaluated, but this approach is computationally expensive. How do you ensure that considering model randomness adequately simulates the DP case?

5. Given that the authors identify high-risk samples misclassified as low-risk by traditional MIAs, it would be beneficial to include visual examples of these samples to see if they appear as outliers by human perception.

6. How does the gap between the proposed approach and traditional MIAs change with the complexity of the datasets? In Table 1, CIFAR-10 shows the lowest miss rates. Would CIFAR-100 yield even lower miss rates? If so, would the gap between the methods diminish when applied to more complex datasets in practice?

7. What would Figure 3 look like if tested with CIFAR-10 or CIFAR-100 datasets?

---

> ### Author Response · Authors · 2024-11-20
> **Responses to questions 1-5**
>
> We thank the reviewer for their insightful and thorough comments on our paper. We individually answer the raised questions below.
>
> **Figure 2a:**
>
> The dashed line does indeed represent the value 0.53. We apologize for the oversight and thank the reviewer for pointing this out. We will update the figure to include the annotation for the dashed line.
>
> **Privacy risk calculation:**
>
> To calculate the privacy risk of a target record, we use the trained MIA to infer the presence of the target record in the training set of each evaluation model. We then calculate the accuracy and AUC for the MIA, where the predictions are the MIA’s output for each evaluation model for the given record, and the labels are the true memberships of the target record. The AUC is independent of the threshold, and therefore we calculate it directly using the predictions and labels for each target record separately. To calculate accuracy, we determine a threshold for each target record.
>
> **TPR at low FPR**:
>
> In “Membership Inference Attacks from First Principles”, the authors evaluate the MIA across all records in the dataset, while we focus on per-record evaluation. When evaluated across records, TPR at low FPR acts as a dataset-level metric: it allows the data owner to determine the records for which the MIA confidently and correctly infers membership. This makes it highly important for understanding the overall risk of a dataset.
>
> However, in our context, where we train and evaluate an MIA separately for each record, TPR at low FPR would only indicate the specific seeds used to train the evaluation models where the MIA performs well. As the seed is random and not a property of the dataset nor the record, we do not consider this particularly meaningful in this context.
>
> **Model-level and record-level risk:**
>
> In our work, we focus specifically on record-level risk estimation, and we argue that the way record-level risk is evaluated (i.e. the traditional setup), is incorrectly averaging the risk of a record across randomly sampled datasets. To gain an estimate of the risk, some randomness is required, and we argue that model randomness (as in the model-seeded setup) is a better source of randomness than dataset sampling (as in the traditional setup) for estimating the risk of a target record in a target dataset. In our experiments, we show that the model-seeded and traditional (averaged out) risks are indeed different, emphasising the need to use the model-seeded setup to obtain a risk estimate closer to the real-life risk.
>
> **Visualization of misclassified records:**
>
> To gain insight into the level to which the misclassified records are outliers in the target dataset, we perform the following analysis:
> For each record in the target dataset, we calculate an "outlier measure" as defined by Meeus et al. (2023). This measure is determined by the mean cosine distance of a record to its 100 nearest neighbors. For CIFAR10, we computed these distances using the embeddings derived from the output of the last linear layer of a ResNet model trained on the full CIFAR10 training dataset, which consists of 30,000 records.
>
> Figure 7 in Appendix F of the revised version of our paper shows the distribution of the outlier measures of the records in the target dataset, and the distribution of the outlier measures of the misclassified records. The results show that the misclassified records tend to have higher outlier scores. We do note that some misclassified records are less of an outlier according to this measure. This suggests that, while outliers do tend to be particularly vulnerable, being an outlier is not the sole indicator of misclassification, and there are other factors that affect a record’s risk.
>
> We include this discussion and the relevant figures in Appendix F of the revised version of our paper.
>
> References:
>
> *Matthieu Meeus, Florent Guepin, Ana-Maria Cre¸tu, and Yves-Alexandre de Montjoye. Achilles’
> heels: vulnerable record identification in synthetic data publishing. In European Symposium on
> Research in Computer Security, pp. 380–399. Springer, 2023*

---

> > ### Author Response · Authors · 2024-11-20
> > **Responses to questions 6 and 7**
> >
> > **CIFAR-10 and CIFAR-100:**
> >
> > We share the same intuition as the reviewer, and we would expect larger and higher-dimensions datasets to lead to lower miss rates.
> >
> > We run the same experiment as for CIFAR-10 for CIFAR-100, calculating the traditional and model-seeded risks for 80 target records. As CIFAR-100 has more label classes than CIFAR-10, the size of the datasets used to train shadow and evaluation models for CIFAR-10 is not sufficient for obtaining well-performing models on CIFAR-100. We therefore train them on datasets containing 20,000 records (compared to 10,000 for CIFAR-10), to obtain similar performance.
> > We obtain the following results:
> >
> > RMSE(AUC_model_seeded, AUC_traditional) = 0.04
> >
> > RMSE(accuracy_model_seeded, accuracy_traditional) = 0.09
> >
> > miss_rate(metric=AUC, threshold=0.8) = 0.08
> >
> > miss_rate(metric=accuracy, threshold=0.8) = 0.4
> >
> > We do indeed see lower miss rates with a larger dataset size and a higher number of classes, and we would expect them to be even lower for datasets with more features or even more records. In this case, as both the number of classes and the dataset size are larger than for CIFAR-10, we cannot determine exactly the extent to which the factors individually contribute to the lower miss rate. We will therefore conduct an ablation study on CIFAR-10 and include the results in the final version of our paper.
> >
> > **Figure 3 for CIFAR-10:**
> >
> > We include the figure for CIFAR-10 in Appendix E of the revised version of our paper.

---

> > ### Comment · Reviewer_x1tv · 2024-11-27
> >
> > Thanks a lot for the response.  Figure 7 is very interesting, it seems that all samples with high outlier measures i.e., bigger than 0.2 are correctly identified. As for the misclassified sample, do you have an idea on what other factors might be in affecting its risk?  Would it be reasonable to have multiple measures of outliers and then aggregated them for better performance?
> >
> > Regarding the intuition that ' larger and higher-dimensions datasets to lead to lower miss rates', I appreciate the additional result provided by the authors.  Including the ablation study is really necessary for understanding the interplay of different factors.

---

### Meta-Review · Area_Chair_TfzD · 2024-12-22

**Metareview:**

This paper proposes the notion of model-seeded membership inference attack, where the data sampling randomness is fixed and attacker uncertainty primarily arises from model randomness. The authors argue that this setup better captures record-level privacy leakage risk, where outlier samples contribute disproportionally to the model compared to inlier samples.

Reviewers raised several concerns regarding this model-seeded setup, including similarity in setting to prior work by Ye et al. (2022) and unclear practical implication when evaluating MIA under this setup. Significant computational overhead is also needed to evaluate attacks under this setup since an attack model is trained for each record in the training set, which severely limits its deployment. Given these weaknesses, AC believes the paper is not ready for publication in its current form.

**Additional Comments On Reviewer Discussion:**

Reviewers and authors discussed the above raised weaknesses, but ultimately did not reach resolution.

---

### Decision · Program_Chairs · 2025-01-22

Reject